# Learning Linear Non-Gaussian Polytree Models

**Daniele Tramontano**[1]    **Anthea Monod**[2]    **Mathias Drton**[1]

[1]Department of Mathematics and Munich Data Science Institute, Technical University of Munich, Germany
[2]Department of Mathematics, Imperial College London, UK

## Abstract

In the context of graphical causal discovery, we adapt the versatile framework of linear non-Gaussian acyclic models (LiNGAMs) to propose new algorithms to efficiently learn graphs that are polytrees. Our approach combines the Chow–Liu algorithm, which first learns the undirected tree structure, with novel schemes to orient the edges. The orientation schemes assess algebraic relations among moments of the data-generating distribution and are computationally inexpensive. We establish high-dimensional consistency results for our approach and compare different algorithmic versions in numerical experiments.

## 1 INTRODUCTION

Directed acyclic graphs (DAGs) have been extensively used in causal modeling; the nodes of a graph represent the random variables of the model while the edges represent directed causal effects from source to sink. These causal effects of the parent nodes on the children are quantified by structural equations. In this paper, we take up this framework and study the problem of inferring the graphical structure underlying the causal model, given only observational data [Drton and Maathuis, 2017]. Referred to as structure learning or causal discovery, it is a problem that is difficult due to the statistical curse of dimensionality and computational issues. Effective methods, thus, need to exploit restrictions on the random variables, graphical structure, or structural equations to simplify the problem [Pearl et al., 2016, Peters et al., 2017]. Here, we consider a class of tree-structured graphs, together with linear structural equations where the error terms are mutually independent and non-Gaussian.

Specifically, we work in the versatile causal discovery framework of *linear non-Gaussian acyclic models (LiNGAMs)* [Shimizu et al., 2006, Shimizu and Kano, 2008]. LiNGAMs postulate linear structural equations with non-Gaussian noise terms to describe the relationships among observed variables. The non-Gaussianity assumption allows for consistent estimation of the graph encoding the model from observational data alone and for efficient structure learning algorithms [e.g., Shimizu et al., 2011, Hyvärinen and Smith, 2013, Wang and Drton, 2020, Hoyer et al., 2008b]. Since the complexity of the structure learning problem depends directly on the underlying graph, consistency results for causal discovery algorithms often require some restrictions on the graph, particularly, when high-dimensional consistency results are desired. In this context, the subset of DAGs whose underlying skeleton is a tree—a *polytree*—is the most scalable setting, offering low computational complexity whilst retaining model expressiveness [Pearl, 1988]. In this paper, we propose algorithms to learn a polytree underlying a LiNGAM model.

Learning a polytree may be decomposed into two tasks: extracting the skeleton and determining the orientation of the edges [Rebane and Pearl, 1987, Jakobsen et al., 2021]. Recovering the underlying skeleton may be achieved via the *Chow–Liu algorithm* [Chow and Liu, 1968]. Existing methods for edge orientation entail checking conditional independence, which is usually carried out by serial hypothesis testing and impacts computational efficiency. We instead proceed by exploiting recent insights concerning algebraic relations among moments to determine edge orientation [Robeva and Seby, 2021, Améndola et al., 2021, Wiedermann, 2015, Dodge and Rousson, 2001]. The result is an efficient approach that adapts a classical algorithm to recover the core causal tree structure and augments it with a novel algebraic strategy to determine edge orientation. The proposed algorithms learn the polytree from observational data alone, in a far more scalable manner than existing LiNGAM algorithms that learn more general graph structures.

The remainder of the paper is organized as follows. Section 2 sets the background and theory. Section 3 presents our contributions where a general population version and

*Accepted for the 38ᵗʰ Conference on Uncertainty in Artificial Intelligence* (UAI 2022).

three algorithmic scenarios are studied in detail. Corresponding theoretical guarantees for our proposed algorithms are given in Section 4. Results of numerical experiments are presented in Section 5. We close with a discussion and suggestions for future research in Section 6. The proofs of all the results are provided in the Appendix A, which is part of the supplementary material. Appendix B in the supplementary material gives the detailed description of the sample versions of the algorithms considered in the paper.

# 2 LINEAR NON-GAUSSIAN STRUCTURAL CAUSAL MODELS

A directed graph (digraph) is a pair $G = (V, E)$, where $V$ is the set of vertices and $E \subset V \times V$ is the set of directed edges. We let $V = [p] := \{1, \ldots, p\}$. An element $(i, j) \in E$ may also be denoted by $i \to j$. A digraph $G$ is acyclic (i.e., a DAG) if it does not contain any directed cycle: there is no sequence of vertices $i_0, \ldots, i_k$ with $i_j \to i_{j+1} \in E$ for $j = 0, \ldots, k-1$ and $i_0 = i_k$. A path in $G$ is a sequence of vertices $i_0, \ldots, i_k$ such that $i_j \to i_{j+1} \in E$ or $i_{j+1} \to i_j \in E$ for all $j$. It is directed if all the arrows point in the same direction. A *polytree* is a DAG in which there is a unique path between any two vertices.

If $i \to j \in E$, then $i$ is a parent of $j$, and $j$ is a child of $i$. If $G$ contains a directed path from $i$ to $j$, then $i$ is an ancestor of $j$ and $j$ is a descendant of $i$. The sets of parents, children, ancestors, and descendants of $i \in V$ are denoted by $\mathrm{pa}(i), \mathrm{ch}(i), \mathrm{an}(i), \mathrm{de}(i)$, respectively.

Let $X = (X_i)_{i \in [p]}$ be a random vector indexed by the vertices of a DAG $G$. For $A \subset [p]$, let $X_A = (X_i)_{i \in A}$. When $X_A$ is conditionally independent of $X_B$ given $X_C$ for disjoint subsets $A, B, C \subset [p]$, we write $A \perp\!\!\!\perp B \,|\, C$. The joint distribution of $X$ satisfies the local Markov property with respect to $G$ if $\{i\} \perp\!\!\!\perp [p] \setminus (\mathrm{pa}(i) \cup \mathrm{de}(i)) \,|\, \mathrm{pa}(i) \,\forall\, i \in [p]$. The Markov equivalence class of $G$ is the set of all DAGs that encode the same conditional independence relations, i.e., for which the set of distributions satisfying the local Markov property is the same. See Maathuis et al. [2019, Chap. 1] for further details.

The skeleton of a DAG is the undirected graph obtained by replacing each directed edge by an undirected edge. Here, edges are denoted by $\{i, j\} \subseteq E$.

## 2.1 STRUCTURAL EQUATIONS

A structural equation model hypothesizes that every random variable in $X$ is functionally related to its parent variables: $X_i = f_i(X_{\mathrm{pa}(i)}, \varepsilon_i)$, $i \in V$, where the $\varepsilon_i$ are independent noise terms and the $f_i$ are measurable functions. If the $f_i$ are linear, then we obtain a linear structural equation model

(LSEM). An LSEM can be written in matrix form as

$$X = (I - \Lambda)^{-\top}\varepsilon, \qquad (2.1)$$

where $\Lambda = (\lambda_{ij})$ with $\lambda_{ij} \neq 0$ only if $i \to j \in E$. An LSEM constrains the dependence structure on the coordinates of $X$, but not the mean. Hence, when working with the LSEM, we may assume without loss of generality that $\mathbb{E}[\varepsilon_i] = 0$, which implies $\mathbb{E}[X_i] = 0$ for all $i \in V$.

Let $\varepsilon^{(2)} = (\mathbb{E}[\varepsilon_i \varepsilon_j])_{ij}$ be the covariance matrix of $\varepsilon$, which is a diagonal matrix by independence, and write $\varepsilon_i^{(2)} := \mathbb{E}[\varepsilon_i^2] > 0$ for its $i$th diagonal entry. The covariance matrix of $X$ is then the positive definite matrix

$$\Sigma = (I - \Lambda)^{-\top}\varepsilon^{(2)}(I - \Lambda)^{-1}. \qquad (2.2)$$

## 2.2 CUMULANTS IN GAUSSIAN AND NON-GAUSSIAN MODELS

Cumulants are alternative representations of moments of a distribution. Here, we formalize the definition in higher order settings and discuss their implications under Gaussian and non-Gaussian errors.

**Definition 2.1.** *The $k$th cumulant tensor of a random vector $(X_1, \ldots, X_p)$ is the $k$-way tensor in $\mathbb{R}^{p \times \cdots \times p} \equiv (\mathbb{R}^p)^k$ whose entry in position $(i_1, \ldots, i_k)$ is the joint cumulant*

$$\mathrm{cum}(X_{i_1}, \ldots, X_{i_k}) :=$$

$$\sum_{(A_1, \ldots, A_L)} (-1)^{L-1}(L-1)! \mathbb{E}\left[\prod_{j \in A_1} X_j\right] \cdots \mathbb{E}\left[\prod_{j \in A_L} X_j\right],$$

*where the sum is taken over all partitions $(A_1, \ldots, A_L)$ of the multiset $\{i_1, \ldots, i_k\}$.*

In our context, the variables have mean 0, so

$$\mathrm{cum}(X_i) = \mathbb{E}[X_i] = 0,$$
$$\mathrm{cum}(X_{i_1}, X_{i_2}) = \mathrm{Cov}[X_{i_1}, X_{i_2}] = \mathbb{E}[X_{i_1} X_{i_2}].$$

More generally, the sum can be restricted to the partitions in which all blocks $A_i$ have at least two elements. In particular,

$$\mathrm{cum}(X_{i_1}, X_{i_2}, X_{i_3}) = \mathbb{E}[X_{i_1} X_{i_2} X_{i_3}],$$
$$\mathrm{cum}(X_{i_1}, X_{i_2}, X_{i_3}, X_{i_4}) = \mathbb{E}[X_{i_1} X_{i_2} X_{i_3} X_{i_4}]$$
$$- \mathbb{E}[X_{i_1} X_{i_2}]\mathbb{E}[X_{i_3} X_{i_4}] - \mathbb{E}[X_{i_1} X_{i_3}]\mathbb{E}[X_{i_2} X_{i_4}]$$
$$- \mathbb{E}[X_{i_1} X_{i_4}]\mathbb{E}[X_{i_2} X_{i_3}].$$

The following powerful result dictates a simple condition for Gaussianity of $X$.

**Theorem 2.2.** *[Marcinkiewicz, 1939, Theorem 2] If there exists $k$ such that $\mathrm{cum}(X_{i_1}, .., X_{i_j}) = 0$ for all $j \geq k$, then $k = 3$ and $X$ has a multivariate Gaussian distribution.*

Furthermore, the following results dictate when the assumptions of Theorem 2.2 are satisfied, thus giving rise to Gaussianity, especially under LSEMs.

**Lemma 2.3.** *If the variables $\varepsilon_1, \ldots, \varepsilon_n$ are independent, then $\mathrm{cum}(\varepsilon_{i_1}, \ldots, \varepsilon_{i_k}) = 0$ unless $i_1 = \cdots = i_k$.*

**Lemma 2.4.** *Let the random vector $X$ follow the LSEM from (2.1) with noise vector $\varepsilon$. Let $\mathcal{C}^{(k)}$ and $\varepsilon^{(k)}$ be the kth order cumulant tensors of $X$ and $\varepsilon$, respectively. Then*

$$\mathcal{C}^{(k)} = \varepsilon^{(k)} \bullet \left[(I - \Lambda)^{-1}\right]_{j=1}^{k}$$
$$= \varepsilon^{(k)} \bullet (I - \Lambda)^{-1} \bullet \cdots \bullet (I - \Lambda)^{-1}$$

*is the Tucker product of $\varepsilon^{(k)}$ and $k$ copies of $(I - \Lambda)^{-1}$.*

Notice here that $\mathcal{C}^{(k)}$ reduces to (2.2) when $k = 2$.

See Comon and Jutten [2010] and references therein for proofs of Theorem 2.2 and Lemmas 2.3 and 2.4.

The next definition introduces the cumulant model obtained from the LSEM (2.1).

**Definition 2.5.** *Let $G = (V, E)$ be a DAG, and let $K \geq 2$ be an integer. The Kth cumulant model of $G$ is the set of $K$-way tensors*

$$\mathcal{M}^{(K)}(G) = \{\varepsilon^{(K)} \bullet \left[(I - \Lambda)^{-1}\right]_{j=1}^{K} :$$
$$\Lambda \in \mathbb{R}^E, \ \varepsilon^{(K)} \in (\mathbb{R}^p)^K \text{ diagonal}\}.$$

*Here, $\mathbb{R}^E$ is the set of $p \times p$ matrices with support $E$. Further, the cumulants up to order $K$ defined by $G$ are modeled by*

$$\mathcal{M}^{(\leq K)}(G) = \mathcal{M}^{(2)}(G) \times \cdots \times \mathcal{M}^{(K)}(G). \quad (2.3)$$

By Theorem 2.2, all multivariate Gaussian vectors $X$ correspond to the zero element of $\mathcal{M}^{(K)}(G)$ for $k \geq 3$.

When the errors in an LSEM are Gaussian, all distributional information is captured by the covariance matrix and equivalence issues arise that hinder identifiability of the full graph. It then becomes necessary to consider non-Gaussian settings. Relaxing the constraint of Gaussianity gives rise to the class of LiNGAMs where the underlying graph now becomes identifiable [Shimizu et al., 2006, 2011]. We will exploit this property algorithmically and use the signal provided by higher cumulants; we do this by way of *treks*.

**Definition 2.6** (Multi-Trek). *A $k$-trek between vertices $i_1, \ldots, i_k \in V$ of a DAG $G = (V, E)$ is a collection of directed paths $T = (P_1, \ldots, P_k)$ in $G$ that share the same source and have $i_j$ as the sink of $P_j$ for all $j$. The common source node is the top of the trek $\mathrm{top}(T)$. A trek is simple if the top node is the unique node on all the paths.*

We denote the set of $k$-treks between $i_1, \ldots, i_k$ by $\mathcal{T}(i_1, \ldots, i_k)$ and the set of simple treks by $\mathcal{S}(i_1, \ldots, i_k)$. See Figure 1 for an example.

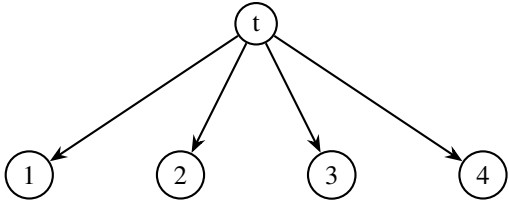

Figure 1: Example of a 4-trek.

If $P$ is a directed path in the DAG $G = (V, E)$ and $\Lambda = (\lambda_{ij}) \in \mathbb{R}^E$, then $\lambda^P = \prod_{(i,j) \in P} \lambda_{ij}$ is a path monomial. For a $k$-trek $T = (P_1, \ldots, P_k)$, set $\lambda^T := \lambda^{P_1} \cdots \lambda^{P_k}$.

**Proposition 2.7** (Multi-Trek Rule). *The kth order cumulant tensor $\mathcal{C}^{(k)}(G)$ of $X$ can be expressed as*

$$\mathcal{C}_{i_1, \ldots, i_k}^{(k)}(G) = \sum \varepsilon_{\mathrm{top}(T)}^{(k)} \lambda^T, \quad (2.4)$$

*where the sum is over all the treks $T$ in $\mathcal{T}(i_1, \ldots, i_k)$ and $\varepsilon_{\mathrm{top}(T)}^{(k)}$ denotes the $\mathrm{top}(T)$ diagonal entry of $\varepsilon^{(k)}$.*

Proposition 2.7 follows from Lemma 2.4 and expanding the entries of $(I - \Lambda)^{-1}$ into sums of path monomials as in the usual trek rule for covariances [Robeva and Seby, 2021].

**Corollary 2.8** (Simple Multi-Trek Rule). *The kth order cumulant tensor $\mathcal{C}^{(k)}(G)$ of $X$ can be expressed as*

$$\mathcal{C}_{i_1, \ldots, i_k}^{(k)}(G) = \sum \mathcal{C}_{\mathrm{top}(S)}^{(k)}(G) \lambda^S, \quad (2.5)$$

*where the sum is extended to all the simple treks $S$ in $\mathcal{S}(i_1, \ldots, i_k)$.*

**Corollary 2.9.** *The ith diagonal entry of $\mathcal{C}^{(k)}$ is*

$$\mathcal{C}_i^{(k)}(G) = \sum_{p_1, \ldots, p_k \in \mathrm{pa}(i)} \lambda_{p_1, i} \cdots \lambda_{p_k, i} \mathcal{C}_{p_1, \ldots, p_k}^{(k)}(G) + \varepsilon_i^{(k)}.$$

## 2.3 POLYTREE MODELS

For general graphs, the algebraic relations among the cumulants may be far more complicated than the bivariate case (which is discussed in Example A.1) and have not yet been fully characterized. However, there exists a generalization of rank-one constraints for polytrees, which we now discuss.

By consequence of there being at most one directed path between any two nodes of a polytree $G$, there is at most one simple trek between any set of nodes $i_1, \ldots, i_k$. The simple multi-trek rule then reduces to $C_{i_1, \ldots, i_k}^{(k)}(G) = \lambda^S C_{\mathrm{top}(S)}^{(k)}$ for a trek between nodes with $S$ being the unique simple trek; denote the top of the simple trek between $i_1, \ldots, i_k$, if it exists by $\mathrm{top}(i_1, \ldots, i_k)$. Also, $C_{i_1, \ldots, i_k}^{(k)}(G) = 0$ if there is no $k$-trek between the nodes.

For any two vertices $i \neq j$, let $c_m^{(i,j),k}$ denote the $k$th order cumulant $\mathcal{C}_{i \ldots i, j \ldots j}^{(k)}(G)$, where the first $m$ indices are equal to $i$ and the remaining $m - k$ equal $j$.

**Proposition 2.10.** *Let $e : i \to j$ be an edge of a polytree $G$. Then the following matrix is of rank one*

$$A^{e,K} = \begin{bmatrix} c_m^{e,k} \\ c_{m-1}^{e,k} \end{bmatrix} \mid 2 \leq m \leq k \leq K \,. \qquad (2.6)$$

The first column of $A^{e,K}$ contains $\mathbb{E}[X_i^2] > 0$. Moreover, for every distribution induced by non-Gaussian errors, there exists $k$ such that $C_i^{(k)} \neq 0$. Hence, at least one minor of $A^{e,K}$ gives us an equation that is satisfied if $i \to j$ is in $G$, and is not satisfied in general for the graph with the edge reversed. This observation will provide the foundation for our learning algorithm, which we now present.

# 3 LEARNING NON-GAUSSIAN POLYTREES FROM MOMENTS

We now present our population algorithm for learning polytrees with three versions for learning the edge orientations. The first common phase is skeleton recovery.

## 3.1 LEARNING THE SKELETON

In its original formulation, the Chow–Liu algorithm gives the maximum likelihood tree approximation of a given discrete distribution [Chow and Liu, 1968]. The tree obtained is the maximum weight spanning tree of the complete undirected graph with edge weights $w(i, j)$, given by the mutual information between $X_i$ and $X_j$. Under a non-degeneracy assumption, the same Chow–Liu algorithm can be used to recover skeletons in the polytree setting [Rebane and Pearl, 1987, Theorem 1]; the proof is based on the following property of the mutual information.

**Proposition 3.1.** *If the polytree that defines the model contains the subgraph $i \to j \to l$ or $i \leftarrow j \to l$, then*

$$\min\{I(X_i, X_j), I(X_j, X_l)\} > I(X_i, X_l),$$

*where $I(\cdot, \cdot)$ is the mutual information.*

When working with an LSEM, a stronger result justifies the use of the absolute value of the correlation coefficient instead of the mutual information.

**Lemma 3.2** (Wright's Formula, [Wright, 1960]). *In the LSEM defined by a polytree, the correlation $\rho_{i,j} = Corr[X_i, X_j]$ satisfies*

$$|\rho_{i,j}| = \begin{cases} \prod |\rho_e|, & \mathcal{T}(i,j) \neq \emptyset, \\ 0, & otherwise, \end{cases} \qquad (3.1)$$

*where the product is taken over the edges of the unique trek from $i$ to $j$, and $\rho_e$ denotes the correlation between the random variables indexed by the endpoints of the edge $e$.*

**Definition 3.3.** *Let $R = (\rho_{i,j})$ be the correlation matrix of a random vector $X$. The Chow–Liu tree $\mathcal{M}(R)$ is the (undirected) maximum weight spanning tree over $[p]$, with weights given by $|\rho_{i,j}|$.*

Kruskal's algorithm may be applied to compute the Chow–Liu tree [Kruskal, 1956].

**Proposition 3.4.** *Let $R = (\rho_{i,j})$ be the correlation matrix of a random vector $X = (X_1, \ldots, X_p)$ that follows the LSEM given by a polytree $G$. If $0 < |\rho_{i,j}| < 1$ for every $e : i \to j \in E$, then $\mathcal{M}(R)$ equals the skeleton of $G$.*

The assumption $|\rho_{i,j}| < 1$ holds for all random vectors with positive definite covariance matrix. Moreover, in a polytree model, $|\rho_{i,j}| > 0$ for an edge $(i, j)$ if $\lambda_{ij} \neq 0$.

## 3.2 LEARNING ORIENTATIONS

We now present three ways to orient the edges in the estimated skeleton. The three resulting orientation algorithms are based on Proposition 2.10 and the following result.

**Theorem 3.5.** *Consider the LSEM given the polytree $G$, and let $e : i \to j$ be an edge of $G$. Then*

*(i)* $\operatorname{rank}(A^{i \to j, K}) = 1$,

*(ii)* $\operatorname{rank}(A^{j \to i, K}) = 2$, *for generic edge coefficients and error cumulants up to order $K$.*

**Proposition 3.6.** *Suppose the skeleton of the polytree $G$ contains the subgraph $i - j - l$ with $\rho_{i,j}, \rho_{j,l} \neq 0$. Then the corresponding subgraph of $G$ is $i \to j \leftarrow l$ iff $\rho_{i,l} = 0$.*

We now present *PairwiseOrientation_Pop*; Algorithm 1. This algorithm takes as input the list of unoriented edges and the parameter $K \geq 3$, which defines the highest order cumulant used in $A^{i \to j, K}$. It orients each edge separately by checking whether the rank of $A^{i \to j, K}$ is 1 or not.

---

**Algorithm 1** PairwiseOrientation_Pop($E, K$)

1: $O \leftarrow \emptyset$
2: **for** $\{i, j\} \in E$ **do**
3:     **if** $\operatorname{rank}(A^{i \to j, K}) = 1$ **then**
4:         $O \leftarrow O \cup \{i \to j\}$
5:     **else**
6:         $O \leftarrow O \cup \{j \to i\}$
    **return** $O$

---

Our second algorithm *TPO_Pop*, Algorithm 2, proceeds recursively. At each step, it takes the order $K$, a list of already oriented edges $O$, a list of still unoriented edges $E$, and, possibly, an oriented edge $o$, as inputs. Here $t(o)$ is the target/sink of the edge and $E \cap t(o)$ is the (possibly empty) set of unoriented edges containing $t(o)$. The procedure checks if there are unoriented edges, and if so, it searches for triplets

of the form $i \to j - k$, where the oriented edge $o = i \to j$ can come either from the previous call of the procedure or from checking the rank of $A^{i \to j, K}$. For such a triplet, the method determines whether $\rho_{i,k} = 0$, orienting the other edge according to the result. The algorithm is initialized with $O = o = \emptyset$ and the full list of undirected edges, $E$.

---

**Algorithm 2** TPO_Pop$(E, K, O, o)$

---

1: **if** $E \neq \emptyset$ **then**
2:    **if** $o = \emptyset$ **then**
3:      $\{i, j\} \leftarrow E[1]$
4:      **if** $\mathrm{rank}(A^{i \to j, K}) = 1$ **then**
5:        $o \leftarrow (i \to j)$
6:        $O \leftarrow O \cup \{o\}$
7:      **else**
8:        $o \leftarrow (j \to i)$
9:        $O \leftarrow O \cup \{o\}$
10:    $E_o \leftarrow E \cap t(o)$
11:    **if** $E_o \neq \emptyset$ **then**
12:      $E \leftarrow E \setminus E_o$
13:      **for** $t(o) - k \in E_o$ **do**
14:        **if** $\rho_{s(o),k} = 0$ **then**
15:          $O \leftarrow O \cup \{k \to t(o)\}$
16:        **else**
17:          $O \leftarrow O \cup \{t(o) \to w\}$
18:          $o \leftarrow (t(o) \to w)$
19:          $O, E \leftarrow TPO\_Pop(E, K, O, o)$
20:    $O, E \leftarrow TPO\_Pop(E, K, O, \emptyset)$
    **return** O,E

---

Our third proposed algorithm *PTO_Pop*, Algorithm 3, can be seen as a direct extension of learning completed partially directed graphs (CPDAG)—a mixed graph that encodes the causal information common to all the members of a Markov equivalence class. Here, we first compute the CPDAG following Rebane and Pearl [1987], then we orient all remaining undirected edges by considering the rank of $A^{i \to j, K}$. This ensures that no other unshielded colliders appear.

The following example compares our three algorithms.

**Example 3.7.** *Consider the graph $G$ with $1 \to 2 \to 3 \leftarrow 4$. With the skeleton $1 - 2 - 3 - 4$ inferred, the algorithm PairwiseOrientation_Pop sequentially computes the rank of $A^{i \to j, K}$ in the order of all edges and orients them according to the results. TPO_Pop orients $1 - 2$ using the rank condition and then checks if $\rho_{1,3} = 0$. Since this is not the case, it orients $2 - 3$ using the rank condition and then $3 - 4$ checking that $\rho_{2,4} = 0$. Finally, PTO_Pop first computes $\rho_{1,3}$ and $\rho_{2,4}$. Since $\rho_{2,4} = 0$, it orients $2 - 3 - 4$, and then orients $1 - 2$ with the rank condition.*

**Theorem 3.8.** *The three versions of the algorithm are correct for generic edge coefficients and cumulants up to order $K$.*

---

**Algorithm 3** PTO_Pop$(E, K)$

---

1: $O \leftarrow \emptyset$
2: **for** $i - j - k \in E$ **do**
3:    **if** $\rho_{i,k} = 0$ **then**
4:      $E \leftarrow E \setminus \{\{i, j\}, \{j, k\}\}$
5:      $O \leftarrow O \cup \{i \to j, k \to j\}$
6: **for** $i \to j \in O$ **do**
7:    **for** $j - l \in E$ **do**
8:      $E \leftarrow E \setminus \{(j, l)\}$
9:      $O \leftarrow O \setminus \{j \to l\}$
10: **for** $\{i, j\} \in E$ **do**
11:    **if** $\mathrm{rank}(A^{i \to j, K}) = 1$ **then**
12:      $O \leftarrow O \cup \{i \to j\}$
13:      **for** $j - l \in E$ **do**
14:        $E \leftarrow E \setminus \{(j, l)\}$
15:        $O \leftarrow O \setminus \{j \to l\}$
16:    **else**
17:      $O \leftarrow O \cup \{j \to i\}$
18:      **for** $i - l \in E$ **do**
19:        $E \leftarrow E \setminus \{(i, l)\}$
20:        $O \leftarrow O \setminus \{i \to l\}$
    **return** O

---

# 4 LEARNING NON-GAUSSIAN POLYTREES FROM DATA

We now consider the empirical versions of our algorithms, which now learn a polytree from a dataset consisting of $n$ i.i.d. random vectors. The algorithms then process the sample correlations $\hat{\rho}_{i,j}$ and sample cumulants $\hat{c}_m^{(i,j),k}$. Let $\hat{\Sigma}_{i,j}$ be the unbiased sample covariances. Then $\hat{\rho}_{i,j} = \hat{\Sigma}_{i,j} / \sqrt{\hat{\Sigma}_{i,i} \hat{\Sigma}_{j,j}}$. Generalizing sample covariances, we take the sample cumulants $\hat{c}_m^{(i,j),k}$ to be the $k$-statistics that estimate $c_m^{(i,j),k}$ in an unbiased manner [McCullagh, 1987, §4.2].

We provide consistency results in a high dimensional setting where the size of the polytree grows at a faster rate than the sample size, subject to log-concavity of the variables. Specifically, we assume the errors $\varepsilon_i$ and thus also the observation vector $X$ are log-concave distributed. This setting allows for the following corollary that builds on the concentration inequality given in Lemma B.3 of Lin et al. [2016].

**Corollary 4.1.** *Let $K \in \mathbb{N}$ and suppose that all moments up to order $2K$ of the random vector $X$ are bounded in magnitude by a constant $M_K > 0$. There exists a constant $L > 0$ such that for any $k \leq K$, if $\hat{c}_m^{(i,j),k} = c_m^{(i,j),k} + \epsilon_m^{(i,j),k}$ is the $k$-statistic for a sample of size $n$, for every $\delta > 0$ where*

$$\frac{2}{LK^2\sqrt{M_K}}\left(\frac{\delta\sqrt{n}}{e}\right)^{\frac{1}{K}} > 2, \text{ we have}$$

$$\mathbb{P}[|\epsilon_m^{(i,j),k}| > \delta] \leq \exp\left\{-\frac{2}{LK^2\sqrt{M_K}}\left(\delta\sqrt{n}\right)^{\frac{1}{K}}\right\}.$$

## 4.1 LEARNING THE SKELETON CONSISTENTLY

Let $\rho_{\min}$ and $\rho_{\max}$ be the respective minimum and maximum of the absolute edge correlations in the set $\{|\rho_{i,j}| : i \to j \in E\}$ with $0 < \rho_{\min}, \rho_{\max} < 1$. We will use the following lemma on the correctness of the Chow–Liu tree $\mathcal{M}(R_n)$ computed from the sample correlation matrix $R_n = (\hat{\rho}_{i,j})$, together with Lemma 7 from Harris and Drton [2013], both restated below.

**Lemma 4.2.** *Let $\gamma = \rho_{\min}(1 - \rho_{\max})/2$. Then the event $F := \bigcap\{|\hat{\rho}_{i,j} - \rho_{i,j}| \leq \gamma\}$ satisfies $F \subset \{\mathcal{M}(\hat{R}_n) = \mathcal{S}(G)\}$.*

**Lemma 4.3.** *If $A, B$ are $2 \times 2$ symmetric matrices, with $A$ positive definite, $a_{1,1}, a_{2,2} \geq 1$, and $||A - B||_\infty < \delta$, then*

$$\left|\frac{a_{1,2}}{\sqrt{a_{1,1}a_{2,2}}} - \frac{b_{1,2}}{\sqrt{b_{1,1}b_{2,2}}}\right| < \frac{2\delta}{1-\delta}. \quad (4.1)$$

We now have the following consistency result for the Chow–Liu tree $\mathcal{M}(R_n)$.

**Theorem 4.4.** *Let $\lambda := \min\{\min_i \Sigma_{i,i}, 1\}$ and let $\gamma$ and $M_2$ be defined as in Lemma 4.2 and Corollary 4.1 respectively. Then*

$$\mathbb{P}(\mathcal{M}(R_n) = \mathcal{S}(G))$$
$$\geq 1 - \frac{3p(p-1)}{2}\exp\left\{-\frac{1}{2L\sqrt{M_2}}\left(\frac{\lambda\gamma\sqrt{n}}{2+\lambda}\right)^{\frac{1}{2}}\right\},$$

*for all $n > \frac{e^2(2+\lambda)^2(4L^2\sqrt{M_2})^4}{\lambda^2\gamma^2}$.*

## 4.2 LEARNING ORIENTATIONS CONSISTENTLY

For every edge $e = \{i, j\}$ in the skeleton $\mathcal{S}(G)$, let $v_r(e), v_w(e) \in \mathbb{R}^{B(K)}$ be the vectors containing the minors of $A^{(r(e)),K}$ and $A^{(w(e)),K}$ involving the first column. Here, $r(e)$ and $w(e)$ are the correct and incorrect orientations of $e$ in $G$, respectively. Let $B(K) = K(K-1)/2 - 1$ be the size of the vectors.

We assume that there exists $\delta > 0$ such that $||v_w(e)|| > \delta$ for all $e \in \mathcal{S}(G)$, where $|| \cdot ||$ is the 2-norm. Let $M_K$, $L$, and $\epsilon_m^{(i,j),k}$ be defined as in Corollary 4.1. Moreover, let $c$ be the vector containing all the cumulants $c_m^{(i,j),k}$ such that edge $\{i,j\} \in \mathcal{S}(G)$ and $0 \leq m \leq k \leq K$. Write $\hat{c}_n$ for the vector containing the sample versions of these cumulants. Finally, let $\epsilon_n$ be the corresponding error vector tracking the differences between the true and sample cumulants.

**Lemma 4.5.** *If $f$ is the difference of two monomials of degree 2 in the variables $c$, then*

$$|f(c + \epsilon_n) - f(c)| \leq 4M_K||\epsilon_n||_\infty + 2||\epsilon_n||_\infty^2. \quad (4.2)$$

For use with data, the proposed algorithms in Section 3 must be modified to allow for sampling variability. In particular, instead of assessing whether or not $\mathrm{rank}(A^{i \to j,K}) = 1$, we check $||\hat{v}_{i \to j}(\{i,j\})|| < ||\hat{v}_{j \to i}(\{i,j\})||$ instead. Here, $\hat{v}$ is the sample analogue of $v$, computed using sample moments. Similarly, for the independence test (vanishing of correlation), we check whether or not the absolute sample correlation is below a threshold $\rho_\theta$; Lemma 4.8 clarifies the possible choices of the threshold. The resulting sample versions of the algorithms are given in Appendix B.

Let $\mathcal{A}_n^{PO}(E, K)$ be the output of Algorithm 1 applied to a sample of size $n$ and let $E_{\mathcal{S}(G)}$ be the edge set of the true skeleton of $G$. Then we have the following consistency result.

**Lemma 4.6.** *Let $\delta' := \min\{\frac{\delta}{4M_K\sqrt{B(K)}}, \frac{\sqrt{\delta}}{\sqrt[4]{4B(K)}}\}$. Then*

$$\mathbb{P}(\mathcal{A}_n^{PO}(E_{\mathcal{S}(G)}, K) = G)$$
$$\geq 1 - 4B(K)(p-1)\exp\left\{-\frac{2}{LK^2\sqrt{M_K}}\left(\delta'\sqrt{n}\right)^{\frac{1}{K}}\right\},$$

*for all $n > \frac{e^2(LK^2\sqrt{M_K})^{2K}}{\delta'^2}$.*

**Theorem 4.7.** *Suppose the data are an $n$-sample drawn from a distribution in the LSEM given by a polytree $G$. Let $\hat{G}$ be the polytree obtained by applying Algorithm 1 to the (undirected) edge set of the Chow–Liu tree $\mathcal{M}(R_n)$. Then $\hat{G} = G$ with probability greater than*

$$1 - 4B(K)(p-1)\exp\left\{-\frac{2}{LK^2\sqrt{M_K}}\left(\delta'\sqrt{n}\right)^{\frac{1}{K}}\right\}$$
$$-\frac{3p(p-1)}{2}\exp\left\{-\frac{1}{2L\sqrt{M_2}}\left(\frac{\lambda\gamma\sqrt{n}}{2+\lambda}\right)^{\frac{1}{2}}\right\},$$

*for all $n > \max\left\{\frac{e^2(2+\lambda)^2(4L^2\sqrt{M_2})^4}{\lambda^2\gamma^2}, \frac{e^2(LK^2\sqrt{M_K})^{2K}}{\delta'^2}\right\}$, with constants defined in Lemma 4.6 and Theorem 4.4.*

**Lemma 4.8.** *Let $\tilde{\gamma} = \min\{\rho_{\min}/3, (1 - \rho_{\max})/2\}\rho_{\min}$, and let $\lambda$ and $M_2$ be as in Theorem 4.4. If $\tilde{\gamma} < \rho_\theta < \rho_{\min}^2 - \tilde{\gamma}$, then the probability that all independence tests carried out by Algorithm 3 yield correct decisions is bounded from below by*

$$1 - \frac{3p(p-1)}{2}\exp\left\{-\frac{1}{2L\sqrt{M_2}}\left(\frac{\lambda\tilde{\gamma}\sqrt{n}}{2+\lambda}\right)^{\frac{1}{2}}\right\},$$

*for all $n > \frac{e^2(2+\lambda)^2(4L^2\sqrt{M_2})^4}{\lambda^2\tilde{\gamma}^2}$. The same statement holds for Algorithm 2.*

**Theorem 4.9.** *Suppose the data are an $n$-sample drawn from a distribution in the LSEM given by a polytree $G$. Let $\hat{G}$ be the polytree obtained by applying Algorithm 3 or 2 to the (undirected) edge set of the Chow–Liu tree $\mathcal{M}(R_n)$. If the threshold satisfies $\tilde{\gamma} < \rho_\theta < \rho_{\min}^2 - \tilde{\gamma}$, then there exists $\alpha^* < p - 1$ such that $\hat{G} = G$ with probability greater than*

$$
1 - 4B(K)\alpha^* \exp\left\{ -\frac{2}{LK^2\sqrt{M_K}}\left(\delta'\sqrt{n}\right)^{\frac{1}{K}} \right\}
$$
$$
- \frac{3p(p-1)}{2}\exp\left\{ -\frac{1}{2L\sqrt{M_2}}\left(\frac{\lambda\tilde{\gamma}\sqrt{n}}{2+\lambda}\right)^{\frac{1}{2}} \right\},
$$

*for all $n > \max\left\{ \frac{e^2(2+\lambda)^2(4L^2\sqrt{M_2})^4}{\lambda^2\tilde{\gamma}^2}, \frac{e^2(LK^2\sqrt{M_K})^{2K}}{\delta'^2} \right\}$.*

**Computational Complexity.** The complexity of the three algorithms is dominated by the $\mathcal{O}(p^2\log(p))$ cost of the Kruskal algorithm which computes the Chow–Liu tree, see [Cormen et al., 2001, Chapter VI]. In terms of the edge orientation, Algorithms 1 have linear computational complexity both in $p$ and $n$ which is independent of the structure of the graph, while Algorithms 3 and 2 may entail a cost that is quadratic in $p$ in the worst case scenario, e.g., a star tree with all the edges outgoing from the center.

## 5 NUMERICAL EXPERIMENTS

We assess and compare the accuracy of our three proposed algorithms on synthetic data, simulated as follows: For any fixed choice of $n$, $p$, and error distribution, we first generate a random undirected tree with $p$ nodes using randomly generated Prüfer sequences [Prüfer, 1918] and then independently orient each edge. Next, we draw $n$ samples for every node from the error distribution and uniformly draw the coefficients $\lambda_{ij}$ from the interval $(-1, -0.3) \cup (0.3, 1)$. Finally, we multiply the matrix of sampled errors by the matrix $(I - \Lambda)^{-1}$ to obtain samples corresponding to the LSEM defined by the generated polytree.

The performance is measured by the structural Hamming distance, which is the number of incorrectly included edges plus the number of incorrectly omitted edges, plus the number of incorrectly oriented edges, divided by $2(p - 1)$. Small distance indicates improved performance. We show the results in three settings: (i) low dimensional, with $p \leq 200$ and $1 \leq n/p \leq 100$; (ii) high dimensional, with $1500 \leq p \leq 3000$ and $0.5 \leq n/p \leq 1$; and (iii) a large scale setting with $10000 \leq p \leq 20000$ and $n/p = 0.1$. We set up experiments with errors drawn from the gamma and uniform distributions; the results are displayed in Figure 2.

For the choice of threshold required in Algorithms 3 and 2, we evaluate the algorithms on a grid of thresholds and report the value corresponding to the best result.

**Gamma Distribution.** Errors were drawn from $\Gamma(\alpha, \beta)$; the shape $\alpha$ and the scale $\beta$ parameters are uniformly drawn

from $(0.5, 5)$. Since $\Gamma(\alpha, \beta)$ is asymmetric, we tested the algorithms with $K = 3$.

The experimental results are coherent with our developed theory: for all three algorithms, the distance between the true and learned trees converges to 0 as the sample size and/or the dimension of the tree increases. We observe that Algorithm 1 performs better both in mean accuracy and variance, despite heavily relying on higher moments which is statistically disadvantageous. The improved performance is potentially due to the fact that Algorithm 1 avoids any potential error propagation since the edges are oriented independently.

**Uniform Distribution.** Errors were drawn from $U(a, b)$, with the parameter $a$ uniformly drawn from $(-10, -1)$ and $b$ uniformly drawn from $(1, 10)$. Here, the uniform distribution is symmetric so the third order cumulants are 0; we thus tested the algorithms with $K = 4$.

The experimental results here are also consistent with our developed theory. We also notice that overall, the experiments with uniform errors outperform those with gamma errors, which may be due to the greater higher order moments associated with the gamma distribution, which tend to increase the variance of the sample cumulants in Corollary 4.1.

The code to reproduce the experiments is available at https://github.com/danieletramontano/LiNGAM-Polytree-Learning.

## 6 CONCLUSION

In this paper, we proposed three algorithms that learn linear non-Gaussian polytree models first using the Chow–Liu algorithm to infer the graph skeleton, and then subsequently applying different approaches to orient edges leveraging non-Gaussianity and marginal uncorrelatedness. The algorithms differ from one another in how much information is taken from correlations versus from higher moments. The numerical experiments show that the algorithms also perform well in very high-dimensional problems. These results indicate that our approach may be applicable in preliminary data analyses towards the aim of understanding dependence structures in data, particularly since the polytree setting allows for richer dependence and causal structures than other tree-based models [e.g., Edwards et al., 2010].

Our work motivates the following questions for future research:

*How to avoid Chow–Liu?* As pointed out above the main computational burden comes from the computation of the Chow–Liu tree. Another shortcoming of the Chow–Liu algorithm is that it requires the whole covariance matrix to be computed and stored beforehand, making it impractical for very large graphs. A solution to this problem that leverages on algebraic relations has been proposed by Lugosi et al. [2021] for undirected trees. A possible extension of this

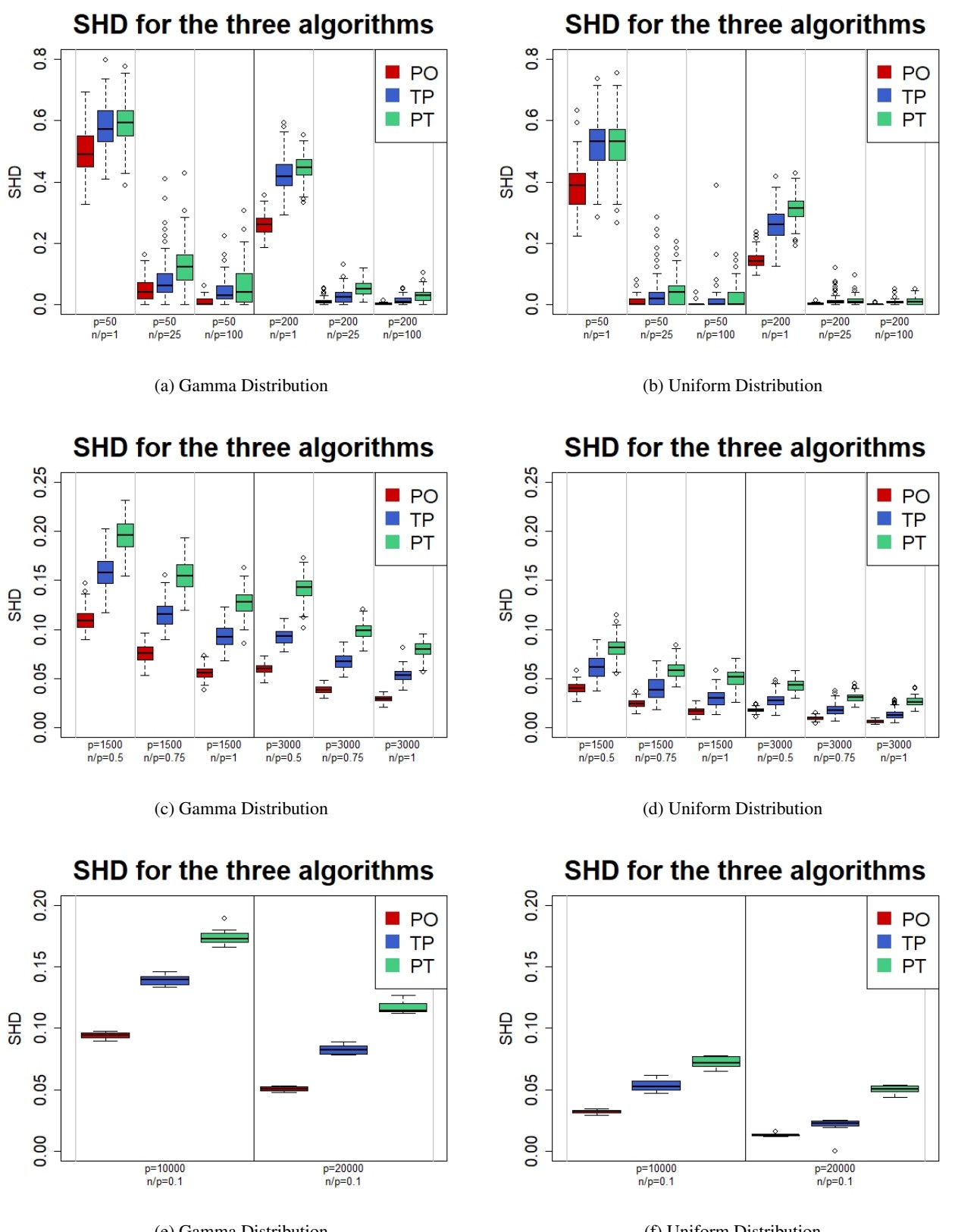

Figure 2: Performance for low dimensional (2a, 2b), high dimensional (2c, 2d) and large scale experiment (2e, 2f) over 200, 100 and 10 runs, respectively.

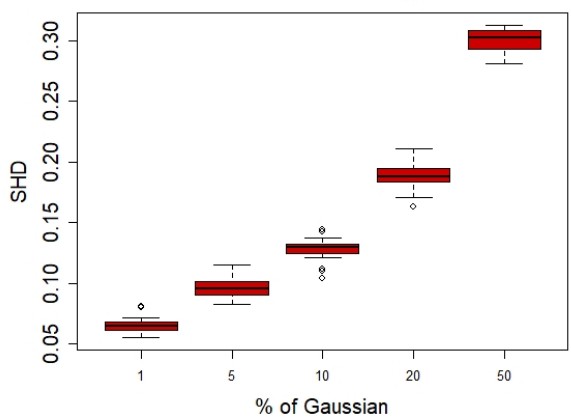

## SHD for the PO algorithm

Figure 3: Performance of Algorithm 1 for high dimensional experiments with varying % of Gaussian random variables over 25 runs.

approach to polytrees would be of interest.

*How to best handle Gaussian random variables when learning a polytree?* In some settings we may encounter the situation that some but not all errors are non-Gaussian; see Hoyer et al. [2008a] for a characterization of equivalence of graphs in this case. An interesting problem is then to determine how the respective performance of our algorithms is affected by partial Gaussianity and provide modifications that effectively learn a polytree equivalence class. As an illustration, Figure 3 shows that Algorithm 1 achieves 70% accuracy when a random choice of half of the random variables are allowed to be Gaussian.

*Which tree structures are the most difficult to learn?* Tan et al. [2009] show that for undirected Gaussian tree models, the star and the chain represent the most difficult and the easiest trees to learn, respectively. The difficulty is due to the correlation decay. An interesting question to pursue is what the polytree analogues for the most difficult and easiest trees to learn would be.

*What happens when the graph is not a tree?* Boix-Adserà et al. [2021] prove a weakness result of the Chow–Liu algorithm under model misspecification for the Ising model and adapt it to achieve a form of optimality. It would be of interest to describe a similar optimality criterion in the LiNGAM setting and investigate how our algorithm performs under these terms.

## Acknowledgements

This project has received funding from the European Research Council (ERC) under the European Union's Horizon 2020 research and innovation programme (grant agreement No 883818). DT's PhD scholarship is funded by the IGSSE/TUM-GS via a Technical University of Munich–Imperial College London Joint Academy of Doctoral Studies (JADS) award (2021 cohort, PIs Drton/Monod).

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
