# OpenReview forum: "Learning Linear Non-Gaussian Polytree Models"
_auai.org/UAI/2022/Conference — UAI 2022 Poster_

### Official Review · Reviewer_qddf · 2022-03-30

**Q2(1) Originality/Novelty:** 3
**Q2(2) Significance/Impact:** 2
**Q2(3) Correctness/Technical Quality:** 3
**Q2(6) Clarity Of Writing:** 3
**Q6 Overall Score:** 5
**Q8 Confidence In Your Score:** 3

**Q1 Summary And Contributions:**

The authors propose several algorithms for learning a polytree underlying a LiNGAM model. The approach is based on the Chow-Liu algorithm for finding the skeleton and then assessing the properties of cumulants for directing the edges. The authors prove consistency and empirically prove the approach in a simulation study.

**Q2 Assessment Of The Paper:**

More detailed information regarding each of these aspects is given below:

**Q2(4) Quality Of Experiments (Optional):**

3: Good: The experimental evaluation is adequate, and the results convincingly support the main claims.

**Q2(5) Reproducibility:**

3: Good: Key resources (e.g., proofs, code, data) are available and key details (e.g., proofs, experimental setup) are sufficiently well-described for competent researchers to confidently reproduce the main results.

**Q3 Main Strengths:**

•	novel ideas on using cumulants for directing edges
•	proof on consistency
•	empirical validation in simulation study


**Q4 Main Weakness:**

•	quite technical; didactical improvement necessary
•	unclear how method compares to competitors


**Q5 Detailed Comments To The Authors:**

Major issues:
•	While the numerical experiments empirically validate the new approach, I am wondering how it compares to already existing methods (phps for general DAGs) in the field both in terms of accuracy and speed. One would expect that the new method is superior if the simulation is restricted to polytrees. Can this be confirmed in a simulation study ?
•	Could the approach for finding orientations easily transferred to the more general DAG setting (I understand that extending the Chow-Liu part is more tricky)? Would that in itself phps improve existing methods for sturcutre learning and if yes, in what situations would you expect most benefits ?
•	Notation and some concepts used are quite intricate. I suggest to use (phps on running) simple example to illustrate the concepts. Especially the concept of the cumulant tensor, the multi-trek and matrix in eq. (2.6) might benifit from further illustrations.
Minor issues:
•	p1: depends directly ON the underlying graph
•	p.2: It is IS directed
•	p.3: use the THE signal
•	p.3: Sentence is not complete: "A trek is simple if the top node the unique node on all the paths"
•	p.3: Please define "path monomial" more precisely, ideally with a formula instead of words
•	p.3: noes (instead of nodes)
•	p.5: In def. of rho_hat: Hat's on Sigmas are missing
•	p.5, Cor. 4.1: What is "L" ? Phps I missed it, but it is hard to keep track of all the constants - sorry
•	general: If you restate Lemmas etc., I suggest you show the reference to the original paper at the start of the Lemma to make the distinction to your original Lemmas etc. more prominent.


**Q7 Justification For Your Score:**

•	some interesting methodological ideas
•	restricted to a very special and rather narrow field of application since restricted to polytrees
•	quite technical, not so easy to read for people not familiar with the field


**Q9 Complying With Reviewing Instructions:**

1: Yes.

---

### Official Review · Reviewer_xyPk · 2022-04-11

**Q2(1) Originality/Novelty:** 3
**Q2(2) Significance/Impact:** 2
**Q2(3) Correctness/Technical Quality:** 3
**Q2(6) Clarity Of Writing:** 2
**Q6 Overall Score:** 4
**Q8 Confidence In Your Score:** 3

**Q1 Summary And Contributions:**

The paper studies causal discovery for LiNGAM models, restricted to polytree structures. It applies the chow-liu algorithm for finding the skeleton and proposes three new algorithms for orienting the edges in the graph. Consistency results for the sample versions of the algorithms are given. An empirical evaluation on simulated data is carried out.

**Q2 Assessment Of The Paper:**

More detailed information regarding each of these aspects is given below:

**Q2(4) Quality Of Experiments (Optional):**

2: Fair: The experimental evaluation is weak: important baselines are missing, or the results do not adequately support the main claims.

**Q2(5) Reproducibility:**

3: Good: Key resources (e.g., proofs, code, data) are available and key details (e.g., proofs, experimental setup) are sufficiently well-described for competent researchers to confidently reproduce the main results.

**Q3 Main Strengths:**

Causal discovery an important important problem, the restriction to the LiNGAM model (for identifiability) and polytrees (for efficiency) makes sense. The presented algorithms seem to work correctly, at least in the low-dimensional setting they indeed recover the generated causal structure when sample size is large enough in the relation to the number of variables.

**Q4 Main Weakness:**

(1) The paper is relatively inaccessible. Good examples for the presented concepts and well as a verbal explanation of the theoretical results could have been useful.

(2) Experiments are not particularly insightful, there is
- only simulated data
- no running time comparisons among the algorithm variants
- no comparison to other causal discovery methods.
While the first point may be justifiable, the other two are problematic, since the abstract claims "significant efficiency over current algorithms", yet there is no empirical evidence for the claim given in the paper.

(3) It is not clear, whether all three algorithmic variants are actually needed, as they perform relatively similar. And even if it's a bit hard to judge due to presentation issue, it seems that the relatively simple Algorithm 4 a.k.a. "Pairwise Orientation", works best.

**Q5 Detailed Comments To The Authors:**

Regarding the following two statements:
"For choice of threshold required in Algorithms 5 and 6, we evaluate the algorithms on a grid of thresholds and report the value corresponding to the best result."
and
"The improved performance [of Algorithm 4] is potentially due to the tuning-free nature of this algorithm compared to the other where tuning of the threshold parameter is required. "
I cannot follow the logic. Shouldn't we expect Algorithm 5 and 6 to perform better when they have a hyperparameter, which is used to cherry-pick the best result, instead of being tuned on data not used for final evaluation?

Presentation:
- The individual plots in Fig. 2-4 are tiny. Please increase size, at expense of number or location of plots [appendix].
- It would also be good to have the results of the different algorithms side-by-side within the same plot (e.g. using different colors). It would make it easier to judge differences among them.
- Be consistent when referring to algorithms: The text uses e.g. "Algorithm 4", whereas the plot is labeled with "P.O."
- typo in Section 2.3 noes -> nodes


**Q7 Justification For Your Score:**

While the focus of the paper is obviously on the theoretical aspects of the problem, I nevertheless believe that a comparison to alternative approaches (even if they do not explicitly make the polytree assumption) are required.

**Q9 Complying With Reviewing Instructions:**

1: Yes.

---

### Official Review · Reviewer_eG3T · 2022-04-12

**Q2(1) Originality/Novelty:** 4
**Q2(2) Significance/Impact:** 2
**Q2(3) Correctness/Technical Quality:** 3
**Q2(6) Clarity Of Writing:** 4
**Q6 Overall Score:** 6
**Q8 Confidence In Your Score:** 4

**Q1 Summary And Contributions:**

The submission describe a novel method for structural learning of polytree models. The method is base on algebraic conditions on the cumulants of the joint distribution of the random variables. In particular three algorithms are proposed, theoretical results are derived and an extensive simulation study is performed to compare the proposed algorithms.

**Q2 Assessment Of The Paper:**

More detailed information regarding each of these aspects is given below:

**Q2(4) Quality Of Experiments (Optional):**

3: Good: The experimental evaluation is adequate, and the results convincingly support the main claims.

**Q2(5) Reproducibility:**

2: Fair: Key resources (e.g., proofs, code, data) are unavailable but key details (e.g., proof sketches, experimental setup) are sufficiently well-described for an expert to confidently reproduce the main results.

**Q3 Main Strengths:**

- very clear and extremely well written paper
- strong theoretical justifications and results
- novel approach

**Q4 Main Weakness:**

- code and implementation of the proposed algorithms is not provided
- an example with real world data would add value to the work
- the methods are not compared to baselines

**Q5 Detailed Comments To The Authors:**

I enjoyed reading the paper and I think this is a very good submission. Congrats to the authors for such work!

The only major complaint I have is about the lack of comparison with some baselines, e.g.
- Chow-Liu skeleton + a bivariate/pairwise method for edge directions (e.g. ANM assuming non-Gaussianity)
- classical LiNGAM without assuming a polytree model

I am more than willing to update the score if the authors will address this point.

Also please consider sharing the code of the methods and the experiments.

Finally, please change/fix the Figures, I had to use 300% zoom to read them. Some of the Figures (e.g. 4) could benefit from different scales and they could probably be joined together in some cases. Labels and legend should be roughly of the same size as normal text.

**Q7 Justification For Your Score:**

The paper deserve a higher score but I think the lack of comparison with baselines methods in the experiments is a point that authors should address, or justify the lack of it otherwise. I understand that the paper is mainly theoretical but I complete experiment section would be a perfect final addition to an otherwise very strong submission.

**Q9 Complying With Reviewing Instructions:**

1: Yes.

---

### Official Review · Reviewer_pJ4E · 2022-04-12

**Q2(1) Originality/Novelty:** 3
**Q2(2) Significance/Impact:** 2
**Q2(3) Correctness/Technical Quality:** 3
**Q2(6) Clarity Of Writing:** 3
**Q6 Overall Score:** 5
**Q8 Confidence In Your Score:** 4

**Q1 Summary And Contributions:**

The paper presents a new algorithm for learning the causal structure in non-Gaussian linear SEM models whose underlying skeleton is a tree. Causal structure learning is an intractable problem in general, so finding settings in which this can be achieved efficiently is of interest to researchers. The algorithm starts by applying the well-known Chow-Liu algorithm to discover the skeleton of the tree. The authors then use recently-discovered algebraic properties of cumulants to orient each edge.

**Q2 Assessment Of The Paper:**

More detailed information regarding each of these aspects is given below:

**Q2(4) Quality Of Experiments (Optional):**

2: Fair: The experimental evaluation is weak: important baselines are missing, or the results do not adequately support the main claims.

**Q2(5) Reproducibility:**

3: Good: Key resources (e.g., proofs, code, data) are available and key details (e.g., proofs, experimental setup) are sufficiently well-described for competent researchers to confidently reproduce the main results.

**Q3 Main Strengths:**

The paper presents an interesting application of techniques from algebraic statistics, which have so far been mostly of theoretical interest to researchers. This work shows that they could be used for causal inference and lead to relatively simple algorithms that only need to evaluate pairwise relationships between variables, without the need to perform more expensive separation queries known from general causal discovery algorithms.

**Q4 Main Weakness:**

Unfortunately, the authors do not discuss specific existing algorithms for this task and do not compare their algorithms against any other algorithms in terms of accuracy or running time, which makes it hard to judge the importance of the results. The authors should compare the performance of their algorithms to existing methods or explain why such a comparison is impractical or impossible. Some of the details are quite confusing and require the reader to consult previous literature e.g. the notation for the Tucker product.

**Q5 Detailed Comments To The Authors:**


- The relationship between Gaussianity and causal direction has been extensively discussed in the following paper:

Hoyer, Patrik, et al. "Nonlinear causal discovery with additive noise models." Advances in neural information processing systems 21 (2008).

I think the authors should cite this work.

- The authors briefly discuss extensions to non-tree settings. Recently, similar algebraic techniques have been used to learn the structure of such models in the undirected case:

Lugosi, Gabor, et al. "Learning partial correlation graphs and graphical models by covariance queries." Journal of Machine Learning Research 22.203 (2021): 1-41.

I think the authors should cite these works.

- page 2 : "A polytree is a directed tree if all the connecting paths are directed" - This sounds like an impossible definition to satisfy - take a tree with one source with arrows pointing outwards from it. Any path from one leaf to another will feature edges pointing towards any of the two leaves, so they will never point "in the same direction".

- What is L in Corollary 4.1? This needs to be defined.

- Do the algorithms assume every edge can be oriented? It seems the answer is yes, but it's not clear

- Example A.1 - not clear where this example belongs in the main text.

- Proof of Thm 4.4 - shouldn't P(E) be P(F)?

**Q7 Justification For Your Score:**

The main strength of the paper is developing a novel strategy for causal discovery in the context of trees based on algebraic techniques. Expanding our repertoire of techniques for causal discovery is important and the paper achieves that, even though the setting is somewhat limited.

The main weakness is the lack of comparison to previous work, which makes it harder to evaluate the significance of the result, hence the score.

**Q9 Complying With Reviewing Instructions:**

1: Yes.

---

### Decision · Program_Chairs · 2022-05-15

**Decision:**

Accept (Poster)

**Comment:**

Meta Review: This paper proposes a causal discovery method based on higher-order cumulants for cases where the causal structure graphs are polytrees.

The theoretical guarantee was provided. The performance was evaluated by artificial data experiments. It was compared with other relevant methods in the authors’ feedback, not in the submission. The reviewers acknowledged the comparison.

The reviewers found the proposal for polytree cases useful. However, they raised some concerns that their result only provides the theory. It does not provide real-world application examples and does not motivate the proposed method well.

The average rating was 5. One reviewer gave the rating 6.

I also think developing causal discovery algorithms for specialized cases including these polytree cases is interesting. I would suggest the authors improve the paper by reflecting on the reviewers’ comments.
Giving a good (potential) application example that suits the polytree assumption is preferable.

For the authors' references, the following papers discuss higher-order moments to estimate causal direction (of two variables)
Y. Dodge and V. Rousson. On asymmetric properties of the correlation coefficient in the regression setting.
The American Statistician, 55(1): 51--54, 2001.

S. Shimizu and Y. Kano. Use of non-normality in structural equation modeling: Application to direction of causation. Journal of Statistical Planning and Inference, 138: 3483--3491, 2008.

W. Wiedermann. Decisions concerning the direction of effects in linear regression models using fourth central moments. In Dependent Data in Social Sciences Research, pp. 149-169, 2015.